# Responses of a Resistive Soot Sensor to Different Mono-Disperse Soot Aerosols

**DOI:** 10.3390/s19030705

**Published:** 2019-02-09

**Authors:** Adrien Reynaud, Mickaël Leblanc, Stéphane Zinola, Philippe Breuil, Jean-Paul Viricelle

**Affiliations:** 1IFP Énergies Nouvelles, Rond-Point de l’échangeur de Solaize, BP 3, 69 360 Solaize, France; mickael.leblanc@ifpen.fr (M.L.); stephane.zinola@ifpen.fr (S.Z.); 2Mines Saint-Étienne, Univ Lyon, CNRS, UMR 5307 LGF, Centre SPIN, 42100 Saint-Étienne, France; philippe.breuil@mines-stetienne.fr (P.B.); jean-paul.viricelle@mines-stetienne.fr (J.-P.V.)

**Keywords:** soot sensor, aerosol, nanoparticle, on-board diagnostic, electrophoresis

## Abstract

Since 2011, the Euro 5b European standard limits the particle number (PN) emissions in addition to the particulate mass (PM) emissions. New thermal engine equipped vehicles also have to auto-diagnose their own particulate filter (Diesel particulate filter or gasoil particulate filter) using on-board diagnostic (OBD) sensors. Accumulative resistive soot sensors seem to be good candidates for PM measurements. The aim of this study is to bring more comprehension about soot microstructures construction in order to link the response of such a sensor to particle size and concentration. The sensor sensitivity to the particle size has been studied using successively an electrostatic and an aerodynamic classification, showing the same trend.

## 1. Introduction

The soot nanoparticle emissions from thermal engines have a harmful impact on human health. Because of their small size, they deeply penetrate our breathing apparatus to reach our lungs and pulmonary alveoli. Then, they can release the carcinogenic chemical compounds they carry such as polycyclic aromatic hydrocarbons (PAH) or sulfates [1]. In 2011, the Euro 5b standard introduced a restriction regarding particle number (PN) for Diesel vehicles and it has been extended to gasoline vehicles in 2014 by the Euro 6b standard. The on-board diagnostic (OBD) of the particulate filter (DPF or GPF) is also mentioned. Currently, a pressure sensor (also called Delta P sensor) measures the pressure loss induced by the particulate matter accumulation in the particulate filter [2]. However, it seems that this device is not accurate enough to detect the particulate filter leaks [3] which can cause a significant increase of tailpipe emissions. 

Other technologies are currently the object of research works such as the radio-frequency (RF) sensors which are based on the measurement of a RF signal in the DPF cavity [4]. The electric charge sensors are also studied [5,6] and rely on the measurement of the electrical current induced by particles pre-charged by a corona charger. The accumulative soot sensors present several advantages in this context: they are cheap and easy to integrate on the light-duty vehicles exhaust line. Among the accumulative soot sensors, we can distinguish the capacitive and the resistive soot sensors [7]. This study is focused on this last technology. In the past years, several studies showed that resistive soot sensors are a good track toward PN measurement [8,9,10]. For now, a particulate mass concentration threshold can be detected [10,11], but further research is required to move towards a PN threshold detection.

The sensitive area of the resistive soot sensor is made of two interdigitated platinum electrodes generating an electric field. As the soot particles are naturally electrically charged in the Diesel engine exhaust gas [12,13], the deposit of the aerosol on the electrodes is accelerated by electrostatic phenomena [14,15]. The sensor measures the conductance of the soot microstructures that create bridges between the electrodes, creating conductive pathways [10]. The work of Grondin et al. revealed the existence of an optimal value of the polarization tension due to an equilibrium between the deposition rate by electrophoresis and the destruction of the conductive pathways by Joule effect [14,15]. Grondin et al. also investigated the influence of the soot composition and found out that the conductance of the sensitive element of the sensor clearly decreases with an increasing organic fraction [14,15]. Another important mechanism influencing the particle deposit is the thermophoresis and it has been used to improve the collection efficiency [16]. Its negative impact on the deposition rate has also been investigated. A temperature gradient in the vicinity of the sensor can be artificially generated by a heater on the back of the substrate [15] or internally caused by the heating of the deposited soot by Joule effect [17]. Other experimental parameters have been studied such as the particle mass flow [10,11] or the sensor nose design [18,19]. These studies lead to a better understanding of the soot particle deposition mechanisms.

The aim of this study is to improve our understanding of the soot deposition mechanisms on a resistive soot sensor by investigating the impact of a new parameter: the particle size. This topic was first introduced in a previous communication [20]. Indeed, this is a way to affect both the drag force and the electrical charge distribution [12], which modify theoretically the force balance of each particle. Since a Diesel soot aerosol is composed of a wide range of particle diameters, the sensor signal may be influenced by each class of sizes. In order to perform a sensitive analysis of the particle size effect, two experimental set-ups have been used. First, the classification has been performed using a Differential Mobility Analyzer (DMA 3081, TSI, Shoreview, MN, USA). Then, an Aerodynamic Aerosol Classifier (AAC, Cambustion, Cambridge, UK) was used.

## 2. Materials and Methods

### 2.1. Experimental Set-up Overview

The objective of the experimental set-up is to provide to a sensor a mono-disperse aerosol from a poly-disperse soot one. It has to be mentioned that most of the set-up (particle generation, temperature monitoring, sensor monitoring) was presented in a previous work [14]. However, the classification part of this set-up is described step-by-step in the following section.

The test bench (cf. Figure 1) was composed of a mini-CAST (Combustion Aerosol STandard, Jing Ltd., Zollikofen, Switzerland) that generated a poly-disperse soot aerosol with a concentration $C_{1}=2.5\times 10^{8}\;\mathrm{part}./{\mathrm{cm}}^{3}$ at the temperature $T_{1}=70\;{}^{\circ}C$ (1). The soot particles were carried to the diluter by a beveled 10 mm diameter stainless steel tube. Then, the aerosol was diluted 10 times by a clean, dry air in a VKL 10 diluter (Palas GmbH, Karlsruhe, Germany) in order to avoid any condensation and to reach temperature and concentration compatible with the inlet conditions of the DMA ($T_{2}=28\;{}^{\circ}C$ and $C_{2}\sim 10^{7}\;\mathrm{part}./{\mathrm{cm}}^{3}$) (2). The classification was operated by either the DMA or the AAC ($T_{3}=28\;{}^{\circ}C$ and $C_{3}\sim 10^{4}-10^{5}\;\mathrm{part}./{\mathrm{cm}}^{3}$ depending on the classification technique) (3).

Finally, the mono-disperse aerosol was driven to the 9 mm internal diameter soot sensor chamber (cf. Figure 2)—heated by a tubular furnace at the temperature of $T_{4}\;=\;180\;{}^{\circ}C$—leading to a Reynolds number Re=55 (4). The flow was drained by a GilAir pump at 1.5 L/min. A simple heat exchanger made of an 1/4 inch external diameter and 50 cm long stainless steel tube was used to decrease the temperature of the heated gas to the ambient temperature. Moreover, a High Efficiency Particulate Air (HEPA) filter was used to protect the pump from potential degradation caused by the soot aerosol.

### 2.2. Resistive Soot Sensor

In a previous study (Ciclamen 2 French national project [21]), a resistive soot sensor has been developed by three partners (École des Mines de Saint-Étienne, IFP Énergies Nouvelles and EFI Automotive). The sensor is composed of an alumina substrate (50 mm × 5 mm × 1 mm) that supports the sensing element (3 mm× 3 mm) on one side and the heater on the other side. They are both made of platinum. The sensing element is composed of two interdigitated electrodes which are made from a platinum ink deposition at the substrate surface by screen-printing. Then, the platinum layer is engraved using a laser [10]. The electrodes are 40 µm width and the inter-electrode width is 20 µm (cf. Figure 3). As well as the electrodes, a heater is printed on the back of the alumina substrate in order to perform active regenerations of the soot to clean up the sensor (700 °C).

The soot sensor response is the conductance of the sensing element. A 6430 Sub-Femtoamp Remote Sourcemeter (Keithley, Solon, OH, United States) is used to apply a polarization tension between the electrodes. It helps to the construction of bridge-like structures between the electrodes (as it can be seen on Figure 3) which are responsible of the conductance increase. The construction time of the first bridge linking the electrodes is called “percolation time”. Both the heater and the sensing element were monitored by a specific LabView software routine developed during a previous study [15].

### 2.3. Soot Generation

In this work, soot particles generated by a burner were studied to model the typical soot particles emitted by a Diesel engine in order to obtain more reproducible experimental conditions, instead of using an actual engine. The mini-CAST is based on a propane-air diffusion flame quenched by a nitrogen flow. The air, nitrogen and propane flows can be modified to produce aerosols with different size distributions and chemical natures [22]. The criterion for choosing among the operating points was the concentration and the median mobility diameter of the generated soot particles. Based on the Grondin et al. study, it led to the operating point: $Q_{C3H8}=3.6\;L/\min$, $Q_{air,oxydation}=93\;L/\min$, $Q_{N_{2},mix}=450\;L/\min$, $Q_{N_{2},quench}=9\;L/\min$. The fuel air equivalence ratio of such a combustion was 0.95, the number and mass concentration of the generated aerosol were respectively $2.5\times 10^{8}\;\mathrm{part}./{\mathrm{cm}}^{3}$ and $91\;\mathrm{mg}/m^{3}$ [15]. The chemical nature of the generated aerosol was investigated by Grondin et al. in a previous work [15]. Several settings have been characterized but for the operating point considered in the present work, it can be highlighted that the carbon black concentration is $70\;\mathrm{mg}/m^{3}$, which represents 77% of the mass concentration of the aerosol. The typical soot particles emitted from Diesel engines are reported to contain a lower proportion of carbon black. The proportion of elemental carbon (mostly carbon black) is most of the time under 70% [23].

Finally, the particle size distribution was measured both with an electrostatic classifier and an aerodynamic classifier as mentioned before, and shown on Figure 4. It can be seen that the median mobility diameter and the median aerodynamic diameter were different (respectively 70 nm and roughly 60 nm) as well as the standard deviation. This can be explained by the relation between the aerodynamic diameter $d_{a}$ and the electric mobility diameter $d_{m}$ which is not linear because of the Cunningham slip correction factor $C_{c}$. Indeed, the relation between the mobility diameter, the volume equivalent diameter and the aerodynamic diameter is given by McMurry et al. [24]:(1)$$d_{a}^{2}C_{c}\left(d_{a}\right)=\frac{1}{\chi }\frac{\rho_{p}}{\rho_{0}}d_{ve}^{2}C_{c}\left(d_{ve}\right)=\frac{\rho_{eff}}{\rho_{0}}d_{m}^{2}C_{c}\left(d_{m}\right),$$
where χ is the shape factor used to correct the drag of non-spherical particles, $\rho_{p}$ the density of the particle, $\rho_{0}$ an arbitrary density (1000 kg/m^3^) and $\rho_{eff}$ the effective density, defined here as the ratio: $\rho_{p}\frac{d_{ve}^{3}}{d_{m}^{3}}$ [24].

### 2.4. Electrostatic Classification by a DMA

Inside the DMA, soot particles are carried between two concentric electrodes by an air flow (sheath flow $Q_{sheath}=15\;L/\min$ in this work). The particles are then submitted to both the Coulomb force and the drag force induced by the sheath flow. By changing the electric field or the sheath flow, the electrical mobility diameter of the outlet mono-disperse aerosol can be modified. After their neutralization by a RX-neutralizer (TSI 3088) it is assumed that the number $N_{n}$ of elementary charges e ($=4.8\times 10^{-10}\;e.s.u.$) carried by any particle follows the Boltzmann equilibrium distribution [25]:(2)$$N_{n}=N_{0}\times exp\left(-\frac{n²e²}{d_{m}k_{B}T}\right),$$
where $N_{0}$ is the number of neutral particles, $d_{m}$ the mobility diameter of the particles, $k_{B}$ the Boltzmann constant and T the temperature of the particles. Consequently, a low number of particles are actually carrying electric charges. Thus, according to equation (2), a small amount only of classified particles are transmitted to the outlet of the DMA i.e., approximately 20% for the median mobility diameter of $d_{m}=70\;\mathrm{nm}$ [26] (cf. Figure 5). Furthermore, the aerosol thus generated is essentially composed of single charge nanoparticles.

### 2.5. Aerodynamic Classification by an AAC

The aerodynamic classification was performed by an AAC (Cambustion). Because of their injection between two rotating cylinders, soot particles are submitted to both an aerodynamic flow (sheath flow) and their own centrifugal force. Thus, particles are classified according to their aerodynamic diameter $d_{a}$ which is the diameter of a sphere with standard density that settles at the same terminal velocity as the particle of interest. Since this method does not classify them according to their charge, a higher number of particles is transmitted to the outlet mono-disperse aerosol (cf. Figure 6). Moreover, we could suppose that the global charge of the aerosol has an impact on its sensitivity to electrophoresis, leading to different deposit phenomena, probably more realistic than the previous technique.

The resolution $R_{s}$ of the AAC is defined as the set point divided by the full width half maximum of the instrument’s transfer function in the size domain [27]. It can be expressed as a function of Δ which represents the width of the size distribution function at mid concentration (cf. Equation (3)):(3)$$R_{s}=\frac{d_{a}}{\Delta },$$

In this work, the value of Δ was kept approximatively constant. The settings are summed up in Table 1.

### 2.6. Experimental Protocol

Before the signal acquisition, one hour was spent heating up the soot sensor chamber, flowing clean and heated air through it. Moreover, the soot generator (mini-CAST) required at least 50 min to reach a steady state operation. The size distribution function was then measured either by the DMA or the AAC and the CPC in order to check the number concentration and the operating of every device of the set-up. Finally, the soot sensor was cleaned, being wiped with a clean ethanol soaked-cloth. After being positioned with the sensing element facing the aerosol flow, the sensor was regenerated at 700 °C for 3 min.

## 3. Results and Discussion

### 3.1. Response to an Electrostatically Classified Aerosol

In order to perform a sensitivity analysis over the mobility diameter, four different diameters were studied: 60 nm, 70 nm, 80 nm and 100 nm. 60 nm, 70 nm and 80 nm were chosen to have a similar concentration (cf. Figure 4). Several sensor signal acquisitions are reported in Figure 7. Moreover, several polarization tensions were studied (30 V, 50 V, 70 V and 100 V). However, the measurements did not let us draw a conclusion about any correlation between the polarization tension and the percolation time. This was consistent with a previous study [14], concluding that for the present operating point of the soot generator (mini-CAST Jing Ltd.) the polarization tension does not affect the response time of the sensor between 40 V and 90 V.

When compared with the response to high concentration aerosols ($10^{8}\;\mathrm{part}./{\mathrm{cm}}^{3}$) [15], the sensor signals were composed of rare and random conductance steps (cf. Figure 7), due to the soot bridge constructions. Indeed, in the case of typical responses to high concentration aerosols, until 6 events per second were detected for the mini-CAST operating point considered in this study, with a polarization tension equal to 60 V [15]. The randomness is explained by the low PN concentration (from $9.0\times 10^{3}\;\mathrm{part}./{\mathrm{cm}}^{3}$ to $1.3\times 10^{4}\;\mathrm{part}./{\mathrm{cm}}^{3}$ depending on the particle size) which made these events infrequent, leading to a low reproducibility. Concerning the height of the steps, it varied from 0.4 µS (60 nm) to 0.02 µS (100 nm). This could be explained by several phenomena such as different morphologies or chemical natures of the soot bridges.

Moreover, a continuous decrease of the conductance was observed after every step. A hypothesis is that an erosion mechanism contributes to decrease the radius of the bridges, decreasing slowly the conductance. Under this assumption, the energy necessary to activate the erosion is provided by the electrical current crossing the bridge-like structure, producing heat by Joule effect. A similar phenomenon has been observed by Sediako et al. [28]: by heating a single soot particle at low temperature (550–800 °C), they measured the real-time morphological changes within an Environmental Transmission Electron Microscope (ETEM).

According to this hypothesis, it is expected that a similar phenomenon may be observed for real Diesel soot. Indeed, Sharma et al. [29] report that the activation energy for Diesel soot is close to the average value for several carbon black samples. However, at the moment, the electrical and thermal conductivities of the soot are not accurate enough to evaluate the temperature of the bridge and then to conclude about the nature of the erosion mechanisms.

Considering the randomness of the signals and their low reproducibility, the percolation time was the only parameter that was studied in detail here. A correlation had been identified between the percolation time and the mobility diameter (Figure 8). However, the 100 nm aerosols were not considered for this correlation due to the lower PN concentration (roughly half the concentration of the other points because of the Gaussian shaped size distribution function). Indeed, previous studies have shown that decreasing concentrations leads to increasing percolation times [10]. Nevertheless, the percolation time did not follow the Gaussian shape of the size distribution function, meaning that the trend induced by the size of the particles was predominant over the concentration effect in this range. 

### 3.2. Response to an Aerodynamically Classified Aerosol

In the previous section, the low concentration of the mono-disperse aerosols was identified as a randomness factor, having a negative impact on the reproducibility of the results. In order to improve the experiments, the aerosol was classified using an AAC, leading to more concentrated aerosols ($1.3\times 10^{4}\;\mathrm{part}./{\mathrm{cm}}^{3}$ vs. $7.2\times 10^{5}\;\mathrm{part}./{\mathrm{cm}}^{3}$) without electric charge segregation. Two different mono-disperse aerosols were studied: 70 nm and 50 nm of median aerodynamic diameter. These specific diameters were chosen because of the close number concentration (cf. Table 1). In each case, two measurements were done to evaluate the reproducibility of the sensor response. Moreover, the polarization tension was 60 V in each case.

In Figure 9, it can be seen that the sensor signals had a significantly different shape, closer to the typical responses observed in previous studies performed at high concentrations and for poly-disperse aerosols [14]. Nevertheless, a zoom on the signal close to the percolation time showed that the conductance was still made of random steps with a higher frequency (cf. Figure 9b). Several negative steps were observed and could be attributed to bridges blow off [30]. What is more, the conductance decreases after each step seemed to be lower or nonexistent and the height of the steps (approximately 0.01 µS) was significantly lower compared to the response to the DMA classified aerosols. Also, the reproducibility of the signal was improved.

Before coming to conclusions, it is important to mention that we can dismiss the impact of the concentration because in this case, 70 nm aerosols had a lower concentration than 50 nm aerosols. Indeed, on the one hand, a lower concentration tends to increase the percolation time. On the other hand, it was previously observed that a higher diameter tends to decrease the percolation time (cf. Figure 8). Consequently, as the percolation time was lower for the 70 nm aerosols, it should be true *a fortiori* if their concentrations were strictly equal.

## 4. Discussions

It was observed that the percolation time increased when the particle size decreased. The same trend was observed using two different classification techniques: a DMA and an AAC. This means that the sensor is sensitive to the particle size. 

Several phenomena can explain the impact of the particle size on the sensor response. The first hypothesis is a geometric criterion saying that a higher number of small nanoparticles is required to link two 20 µm spaced electrodes. This leads to a lower bridge construction frequency that is to say a lower slope of the signal $\frac{dG}{dt}$, which was indeed observed experimentally in Figure 9.

Moreover, the size of the particles also affects their motion in the neighborhood of the electrodes. In their numerical study, Teike et al., wrote the force balance of a particle close to the electrode as [31]: (4)$$m_{p}\frac{\partial {\vec{v}}_{p}}{\partial t}={\vec{F}}_{e}+{\vec{F}}_{d}+{\vec{F}}_{b},$$
where $m_{p}$ is the mass of the particle and ${\vec{v}}_{p}$ its velocity. The forces can be expressed by:
(5)$$\{\begin{array}{c} {\vec{F}}_{e}=-N_{n}e\vec{E}\\ {\vec{F}}_{d}=\chi \frac{d_{ve}}{C_{c}\left(Kn\right)}3\pi \eta_{f}{\vec{v}}_{r}\\ {\vec{F}}_{b}=\vec{\zeta }\sqrt{\frac{216\nu_{f}k_{b}T_{f}}{\pi \rho_{p}d_{ve}^{5}{\left(\frac{\rho_{p}}{\rho_{f}}\right)}^{5}C_{c}\left(Kn\right)\Delta t}} \end{array},$$
where $\vec{E}$ is the electric field generated by the polarization of the electrodes, $d_{ve}$ the geometric equivalent diameter of the particle, $C_{c}$ the Cunningham slip correction coefficient. $\eta_{f}$, $\nu_{f}$, $T_{f}$ and $\rho_{f}$ are respectively the dynamic viscosity, the kinematic viscosity, the temperature and the density of the gas. Δt is the relaxation time of the particle. The randomness of the Brownian motion is modeled by the vector $\vec{\zeta }$, which is a zero-mean, unit-variance-independent Gaussian random number.

As it can be seen in the Equation (5), the drag force ${\vec{F}}_{d}$ can be approximated by the Stokes’ law modified by a shape factor χ and depends directly on the particle diameter. The Brownian force also depends directly on the particle diameter. The Coulomb force, however, depends indirectly on the particle diameter because the number of elementary charges carried by a soot particle follows the Boltzmann equilibrium (cf. equation (2)) which also depends on the size of the particle. Among those three forces, all of them depends on the size of the particle, leading to the conclusion that the trajectory in the vicinity of the sensor varies according to the size of the particle. We can make the hypothesis that the deposit position of the bigger particles is more suitable for the construction of the bridge-like structures rather than smaller particles.

However, the sensitivity to particle size can be explained by other factors, such as the chemical nature of soot. Indeed, the chemical nature of the soot may also depend on their size, and it is known that the electrical conductivity of the bridge-like structures depends on their chemical nature [15].

## 5. Conclusions

The aim of this study was to investigate the sensitivity of a resistive soot sensor response to the particle size. To do this, a mono-disperse aerosol was first generated by a DMA. The physical principle of the DMA led to low concentration aerosols (<10^5^ part./cm^3^), inducing a low reproducibility of the sensor signal. The AAC, due to its physical principle, was able to generate more concentrated aerosols, which was suitable in the context of this study. Indeed, an increased reproducibility was observed.

The trend observed using each classification processes were found to be identical: smaller particles increased the percolation time. Thus, it was shown that the resistive soot sensor was sensitive to the particle size, even at low concentration. The resistive soot sensor is a complex system, thus there are numerous physical and chemical phenomena leading to the trend highlighted in this work. It can be interpreted as a geometric phenomenon implying that a higher number of small particles is required to link the electrodes, rather than big ones.

Finally, the size of the particles impacts several forces, modifying their trajectories. Present results allow to suppose that the bigger particles are deposited in the more suitable zones to build bridges. A numerical analysis of the electrophoresis could bring information about the deposition process of the particulate matter and about the impact of the particle size over the sensor signal. 

## Figures and Tables

**Figure 1 sensors-19-00705-f001:**
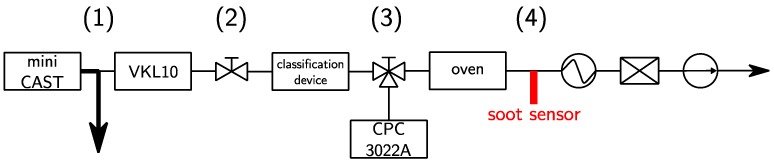
Schematic view of the experimental bench. A CPC 3022A, TSI (Condensation Particle Counter) has been used to measure the distribution size function of the soot aerosol.

**Figure 2 sensors-19-00705-f002:**
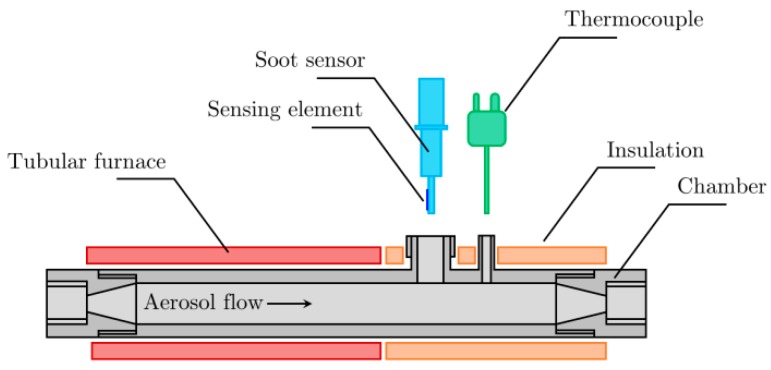
Illustration of the soot sensor chamber.

**Figure 3 sensors-19-00705-f003:**
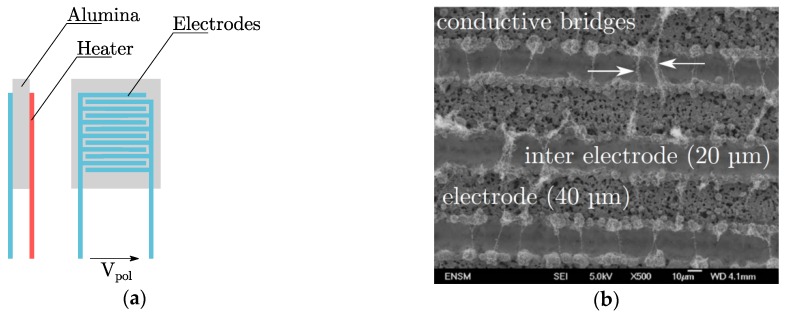
(**a**) Illustration of the resistive soot sensor; (**b**) Typical scanning electron microscopy of the interdigitated electrodes [15]. It shows the soot bridge-like microstructures linking the electrodes after the percolation time is reached.

**Figure 4 sensors-19-00705-f004:**
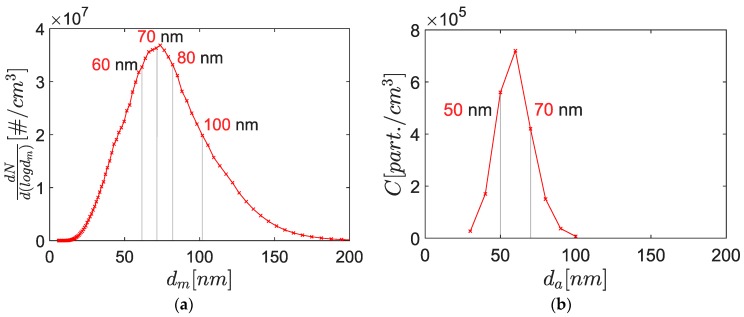
(**a**) Mobility diameter size distribution of a poly-disperse aerosol generated by the mini-CAST. The measurements are performed using a Scanning Mobility Particle Sizer (SMPS); (**b**) Aerodynamic diameter size distribution of the same aerosol measured downstream of an AAC with a CPC.

**Figure 5 sensors-19-00705-f005:**
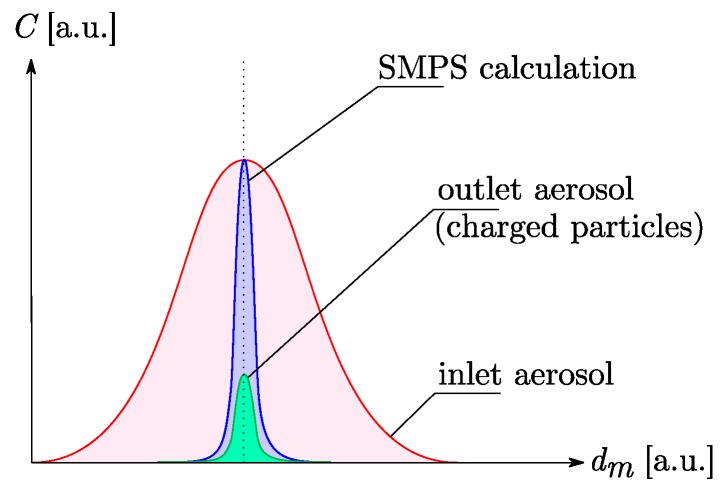
Illustration of the working principle of an aerosol classification by electrostatic technique. The blue area represents the concentration computed by the SMPS algorithm, based on the Boltzmann equilibrium. The effective aerosol (i.e., the particles sent to the sensor chamber) is the green area.

**Figure 6 sensors-19-00705-f006:**
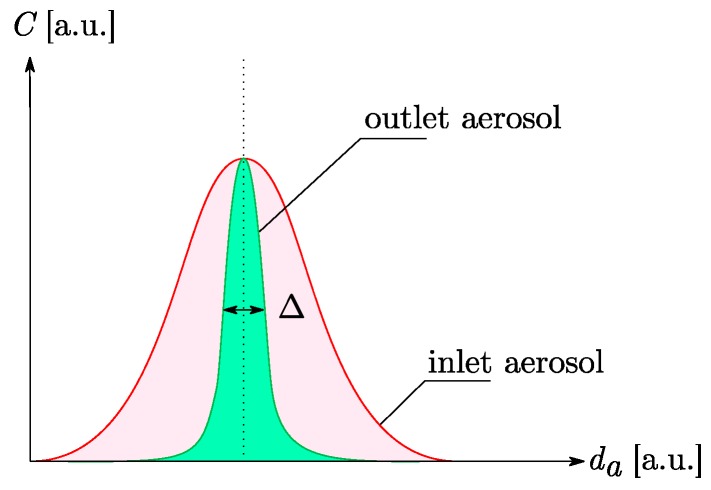
Illustration of the working principle of an aerosol classification by aerodynamic technique.

**Figure 7 sensors-19-00705-f007:**
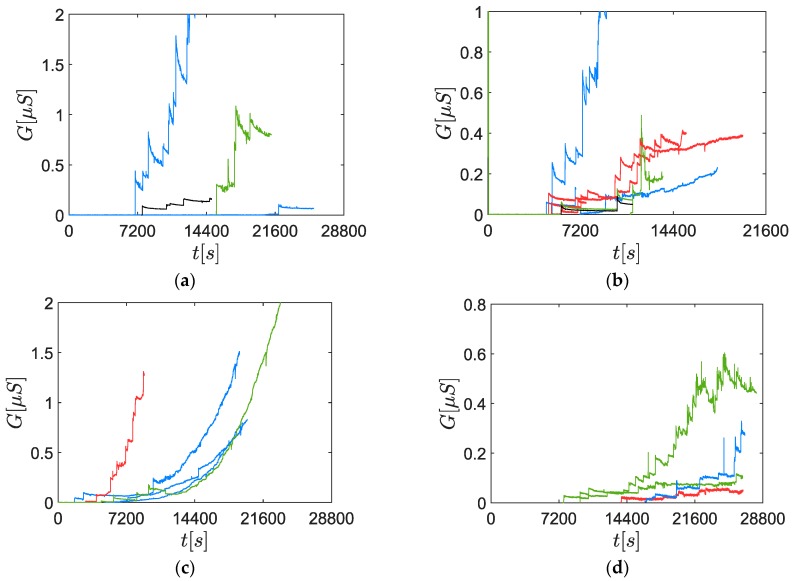
Sample of the soot sensor responses to a DMA-classified aerosol for different mobility diameters and different polarization tensions (red: 30 V; blue: 50 V; green: 70 V; black: 100 V). (**a**) $d_{m}=60\;\mathrm{nm}$; (**b**) $d_{m}=70\;\mathrm{nm}$; (**c**) $d_{m}=80\;\mathrm{nm}$; (**d**) $d_{m}=100\;\mathrm{nm}$.

**Figure 8 sensors-19-00705-f008:**
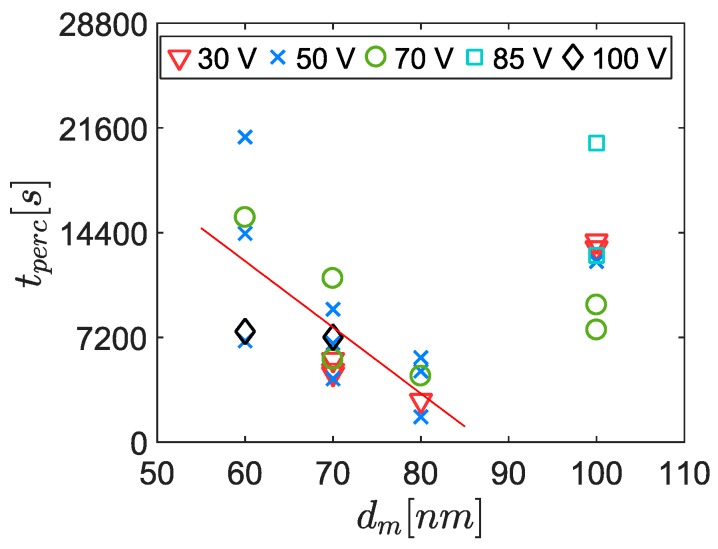
Overview of the percolation time for different mobility diameters and concentrations. The red line is the correlation between the averaged percolation times and the mobility diameters, without taking into account the 100 nm diameter measurements ($t_{perc}=-455\times d_{m}+39737$, $R^{2}=0.9463$).

**Figure 9 sensors-19-00705-f009:**
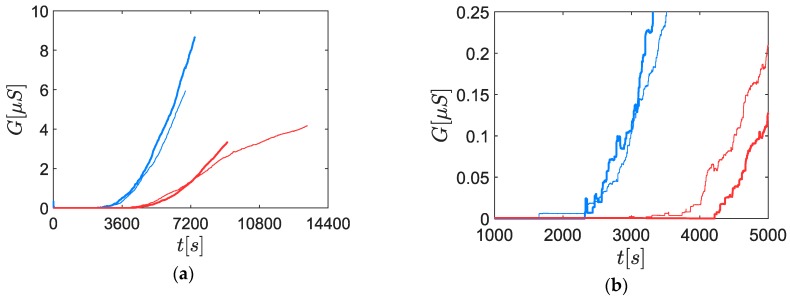
Soot sensor responses of two aerodynamically classified aerosols. Red curves: 50 nm aerosol; blue curves: 70 nm aerosol. The different responses are differentiated by their line width. The polarization tension is $V_{pol}=60\;V$ for each experiment. (**a**) Overview; (**b**) zoom on the percolation time.

**Table 1 sensors-19-00705-t001:** Overview of the AAC parameter $R_{s}$ for different aerodynamic diameters.

*d_a_* [nm]	*R_s_* [-]	Δ [nm]	*C* [part./cm^3^]
50	3.6	13.9	$5.6\times 10^{5}$
70	5.2	13.5	$4.2\times 10^{5}$

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
