# Peer review of "Responses of a Resistive Soot Sensor to Different Mono-Disperse Soot Aerosols"

_sensors, 2019, doi:10.3390/s19030705_

Round 1

Reviewer 1 Report

The manuscript describes a study of the initial response of a resistance soot sensor as a function of aerosol particle size. Both electrostatic and aerodynamic classification of flame generated soot particles is employed in order to obtain size distributions from 50 to 100 nm prior to sensor exposure.

1     The authors make it clear on line 109 that propane combustion is used for soot generation in order to improve reproducibility relative to a diesel engine. While the study is thus of a model system, it may be informative to include some remarks about expected diesel soot behavior with respect to chemical composition. Propane soot is described on line 121 as containing about 80% carbon black which is mostly elemental carbon whereas diesel soot is generally <60% carbon with higher fractions of ash and solvent extractable organics coating particle surfaces.

One aspect is the flow of electricity through the chains. Since electron mobility between carbon particles is related to interfacial properties, the coating layer on diesel soot should have some influence on conductivity along bridges. The other aspect is the bridge erosion by Joule heating mentioned on line 206. Since H N Sharma et al., Experimental Study of Carbon Black and Diesel Engine Soot Oxidation Kinetics Using Thermogravimetric Analysis, Energy Fuels 26 (2012) 5613-5625, report that the activation energy for diesel soot oxidation at 155 kJ/mol is near the average for several carbon black samples it could be concluded that engine soot and propane soot should behave similarly with respect to erosion by Joule heating.

2     If known, it would be of interest to add to the caption in Figure 3b if the image shown is before or after percolation time has been reached.

3     There are two black points for 100 nm in Figure 8 but no black curve in Figure 7d.

4     Conductance steps are described as rare on line 194. This is clear for Figure 7c. They seem to be quite frequent for the other diameters.

5     There is a mixing of present and past tenses when reporting results. Examples in 3.1: line 214 is, 215 was and had, 216 were, 220 does, 221 is. Some effort should be made to achieve consistent verb use, for example all measurements made in the past described with past tense. Same is true for conclusion, for example, verbs in sentences on lines 287 and 288: observed, were, increase, is.

Author Response

1     The authors make it clear on line 109 that propane combustion is used for soot generation in order to improve reproducibility relative to a diesel engine. While the study is thus of a model system, it may be informative to include some remarks about expected diesel soot behavior with respect to chemical composition. Propane soot is described on line 121 as containing about 80% carbon black which is mostly elemental carbon whereas diesel soot is generally <60% carbon with higher fractions of ash and solvent extractable organics coating particle surfaces.

One aspect is the flow of electricity through the chains. Since electron mobility between carbon particles is related to interfacial properties, the coating layer on diesel soot should have some influence on conductivity along bridges. The other aspect is the bridge erosion by Joule heating mentioned on line 206. Since H N Sharma et al., Experimental Study of Carbon Black and Diesel Engine Soot Oxidation Kinetics Using Thermogravimetric Analysis, Energy Fuels 26 (2012) 5613-5625, report that the activation energy for diesel soot oxidation at 155 kJ/mol is near the average for several carbon black samples it could be concluded that engine soot and propane soot should behave similarly with respect to erosion by Joule heating.

Indeed, in this study, the mini-CAST was used in order to model soot emitted from a typical Diesel engine. Concerning the amount of elemental carbon, it was found in the literature that it is highly dependent on the operating condition of the engine [1]. Nevertheless, it seems, indeed, that the EC proportion is under 60% of the total mass most of the time. It was not mentioned in the introduction of the first version of the manuscript, but a previous study investigating the influence of the organic fraction was conducted by Grondin et al. [2]. It was found that the conductance of deposited soot particles was highly dependent on this parameter : the higher the organic fraction, the lower the conductance. Moreover, the sensor behavior was equivalent to the typical sensor response to a Diesel soot aerosol. In conclusion, the impact of the organic coating of soot particles on the response of the sensor was investigated, but the operating point was chosen to optimize the particle number concentration. Indeed, it was necessary to improve the response of the sensor, especially during the DMA classification process.

Considering the bridge erosion, the consideration about the energy activation was added to the manuscript. It allows to extend the behavior of the erosion by Joule heating to other types of soot, if this hypothesis is validated by further studies.

2     If known, it would be of interest to add to the caption in Figure 3b if the image shown is before or after percolation time has been reached.

Indeed, the image of the Figure 3b shows the electrodes of the sensor after the percolation time is reached. It was added to the manuscript.

3     There are two black points for 100 nm in Figure 8 but no black curve in Figure 7d.

Indeed, two curves of the Figure 7d were recorded for a polarization tension of 70 V, and were displayed with the color of a polarization tension of 30 V. This mistake is corrected in the current version of the manuscript.

4     Conductance steps are described as rare on line 194. This is clear for Figure 7c. They seem to be quite frequent for the other diameters.

Depending on the responses on Figure 7a to 7d, the conductance steps happen with a period between 1 minute and 2 hours. The conductance steps are described as rare relatively to the sensor responses to typical polydisperse aerosols generated by the burner mini-CAST. The concentration of such an aerosol is about 2.108 part./cm3 which allows the bridges to be quickly built and to link the electrodes with a higher frequency [3]. Indeed, Grondin et al. [3] show in his PhD thesis that until 6 conductance steps per second were detected for the mini-CAST operating point considered in this study, with a polarization tension equal to 60 V.

 Also, the frequency of the conductance steps may be compared to the responses to the AAC classified aerosols. It can be seen on Figure 9a that the signals are smooth while if it is zoomed in, the Figure 9b shows that the signal is actually made of conductance steps with a period between 10 seconds and 100 seconds (at the beginning of the signals).

Thus, the comparison was clarified in the manuscript. The reference to the PhD thesis containing the analysis of the sensor response showing that it is made of discrete conductance steps happening with a period lower than 1 second was added.

5     There is a mixing of present and past tenses when reporting results. Examples in 3.1: line 214 is, 215 was and had, 216 were, 220 does, 221 is. Some effort should be made to achieve consistent verb use, for example all measurements made in the past described with past tense. Same is true for conclusion, for example, verbs in sentences on lines 287 and 288: observed, were, increase, is.

Thank you for these accurate corrections. It has been taken into account and the tenses were homogenized.

References

[1] S. D. Shah; D. R. Cocker; J. W. Miller; J. M. Norbeck. Emission Rates of Particulate Matter and Elemental and Organic Carbon from In-Use Diesel Engines. Environ. Sci. Technol. 2004, 38, 2544–2550, doi:10.1021/es0350583.

[2] D. Grondin; A. Westermann; P. Breuil; J.-P. Viricelle; P. Vernoux. Influence of key parameters on the response of a resistive soot sensor. Sens. Actuators, B 2016, 236, 1036–1043, doi:10.1016/j.snb.2016.05.049.

[3] D. Grondin. Ph. D., Développement d'un capteur de suie pour application automobile: Etude des paramètres clés affectant sa réponse; Ecole des Mines de Saint-Etienne, Saint-Etienne, 2017.

Reviewer 2 Report

In this paper, a soot particle classification in the context of a resistive soot sensor was studied. The study is interest to readers. However, there are some problems in the manuscript.

1.      Please check the manuscript for linguistic errors.   

2.      Table 1 should be placed around its context.

3.      Figure 7- Figure 9 are invisible to readers and need repainting.

4.      More experiments are necessary for evaluating the approach of the soot particle classification.

I would suggest to revise the paper to make it proper representative of the presented work.

Author Response

1.      Please check the manuscript for linguistic errors.  

The manuscript has been corrected. Emphasis has been placed on tenses to achieve a consistent use. For instance, past tenses have been used for the experiments made in the past.

2.      Table 1 should be placed around its context.

The Table 1 presents the parameters used in order to classify the soot particles emitted by the mini-CAST using the AAC device. Those parameters include, for each studied aerodynamic diameters, the resolution of the monodisperse aerosols, the width of the size distribution at mid concentration. The last column corresponds to the number concentration of the outlet aerosol, once the AAC was set up. The concentrations were measured with a CPC. Could you please precise what could be improved in order to place the Table 1 around its context?

3.      Figure 7- Figure 9 are invisible to readers and need repainting.

Figure 7 to 9 have been repainted with better colors. Moreover, on Figure 9, the curves describing the same experiments have been differentiated using their line width.

4.      More experiments are necessary for evaluating the approach of the soot particle classification.

The objective of the work presented in this manuscript was to evaluate the impact of monodisperse aerosols on a resistive soot sensor response. The median particle size of the monodisperse aerosols varied from 60 nm to 100 nm. Their classification was performed by two different techniques (electrostatic and aerodynamic). This study is considered as a step forward to a better understanding of the deposition mechanisms (diffusion, electrophoresis) of the soot particles on the surface of the sensor electrodes, under the action of an electric field.

It might be unclear in the manuscript, and especially in the title, but the objective was not to demonstrate the capability of the resistive soot sensor to classify nanoparticles, and to provide information about their size. In particular, the term “classification in the context of a resistive soot sensor” induced ambiguity on the aim of this work. That is why the authors suggest that the title should be changed to “Responses of a resistive soot sensor to different monodisperse soot aerosols”.

I would suggest to revise the paper to make it proper representative of the presented work.

Round 2

Reviewer 2 Report

The paper has been improved a lot.The manuscript is now ready for publication.